# Mechanical/Electrical Characterization of ZnO Nanomaterial Based on AFM/Nanomanipulator Embedded in SEM

**DOI:** 10.3390/mi12030248

**Published:** 2021-02-28

**Authors:** Mei Liu, Weilin Su, Xiangzheng Qin, Kai Cheng, Wei Ding, Li Ma, Ze Cui, Jinbo Chen, Jinjun Rao, Hangkong Ouyang, Tao Sun

**Affiliations:** Shanghai Key Laboratory of Intelligent Manufacturing and Robotics, School of Mechatronic Engineering and Automation, Shanghai University, Shanghai 200072, China; mliu@shu.edu.cn (M.L.); SWL19972@163.com (W.S.); wdmzjqxz@163.com (X.Q.); 13771387415@163.com (K.C.); dingwei@shu.edu.cn (W.D.); malian@shu.edu.cn (L.M.); cuize0421@126.com (Z.C.); jbchen@shu.edu.cn (J.C.); jjrao@shu.edu.cn (J.R.)

**Keywords:** nanomanipulator, atomic force microscope, ZnO, mechanical/electrical characterization, piezoelectric property

## Abstract

ZnO nanomaterials have been widely used in micro/nano devices and structure due to special mechanical/electrical properties, and its characterization is still deficient and challenging. In this paper, ZnO nanomaterials, including nanorod and nanowire are characterized by atomic force microscope (AFM) and nanomanipulator embedded in scanning electron microscope (SEM) respectively, which can manipulate and observe simultaneously, and is efficient and cost effective. Surface morphology and mechanical properties were observed by AFM. Results showed that the average Young’s modulus of ZnO nanorods is 1.40 MPa and the average spring rate is 0.08 N/m. Electrical properties were characterized with nanomanipulator, which showed that the ZnO nanomaterial have cut-off characteristics and good schottky contact with the tungsten probes. A two-probe strategy was proposed for piezoelectric property measurement, which is easy to operate and adaptable to multiple nanomaterials. Experiments showed maximum voltage of a single ZnO nanowire is around 0.74 mV. Experiment criteria for ZnO manipulation and characterization were also studied, such as acceleration voltage, operation duration, sample preparation. Our work provides useful references for nanomaterial characterization and also theoretical basis for nanomaterials application.

## 1. Introduction

Nanomaterials have a wide range of applications, whose characterization, including force, electricity, heat and light, etc. is of great significance [1,2,3,4]. For instance, ceramic nanomaterials are often used as friction protective films, so its micro/nano hardness, Young’s modulus, fracture strength and so on need to be tested [5]. Metal or semiconductor nanowires are commonly used as connecting wires or functional construction materials in nano-devices, so that its electrical properties need to be characterized [6]. Performance optimization and reliability improvement of nano-devices rely on nanomaterials.

ZnO is one of the most important nanomaterials in nanotechnology [7]. It is a wide bandgap semiconductor, which has piezoelectric and pyroelectric properties [8,9,10]. With piezoelectric properties, ZnO nanomaterials were used to prepare piezoelectric nanogenerators and nanosensors with different structures to transform mechanical energy into electrical energy [11,12,13]. However, current research on the mechanical/electrical performance of ZnO is still insufficient, and further exploration is still in demand.

SEM and AFM are widely used in basic micro/nano operation and characterization, research of micro/nano science and technology has been continuously innovated and improved [14,15,16]. Nanostructures and nanomaterials with excellent mechanical/electrical properties have been discovered continuously, which has an important impact on the development of life science, information and communication, new materials, and many other fields [17]. AFM force measurement technology provides a technical platform for micro/nano structures mechanical properties characterization [18,19,20]. SEM is able to integrate MEMS equipment and carry out secondary development because of its large vacuum chamber [21]. Most of present mechanical/electrical characterization methods cannot simultaneously measure and observe nanomaterials, limiting work efficiency. The SEM-embedded nanomanipulator and AFM can simultaneously operate and characterize the performance of micro and nano samples.

In this paper, ZnO nanomaterials, including nanorod and nanowire are characterized by AFM and nanomanipulator embedded in SEM respectively. Surface morphology and mechanical properties were observed by AFM. Electrical properties were characterized with nanomanipulator, which showed that the ZnO nanomaterial have cut-off characteristics and good schottky contact with the tungsten probes. Piezoelectric property was also measured. A two-tip approach for piezoelectric measurement was proposed, which is flexible and adaptable. Different with previous publications, piezoelectric properties of a single ZnO nanowire was measured instead of ZnO arrays, which is easy to operate and cost-effective. It is easier to control experimental variables. Experiment criteria for ZnO manipulation and characterization were also studied, such as acceleration voltage, operation duration, sample preparation.

## 2. Theoretical Modeling Analysis

### 2.1. Mechanical/Electrical Characterization Principles

One-dimensional materials such as ZnO nanorods and ZnO nanowires have excellent physical properties due to the size effect, which is an ideal component for constructing nano devices. Research on mechanical properties of materials is helpful to understand internal structure and offer guidance for performance improvement. Nano indentation technology is a widely used method to measure mechanical properties of nanomaterials. Young’s modulus and rigidity are measured with AFM embedded in SEM. Nano indentation technology is based on elastic contact theory. Principle of this technique is to form a force-displacement curve by pressing down sample with a known type of probe and then lifting up the probe. The elastic-plastic properties of nanomaterials were recorded by force-displacement curve.

Electrical characterization of ZnO nanorods were carried out by a nano manipulator embedded in SEM. Two tungsten probes serve as electrodes, which contacts with ZnO nanorod, establishing a conductive circuit together with a power supply. The equivalent electric circuit of metal-semiconductor-metal structure is shown in the Figure 1. There are two probe-ZnO metal semiconductor junctions in the loop. When probes get into contact with ZnO, the forbidden band of semiconductor at the interface bends, forming a Schottky barrier, whose presence leads to a high interfacial resistance [22].

Electrical conductivity is an important parameter to measure the ability to transfer current. Electrical conductivity (σ) can be calculated by Equation (1).
(1)$$R=\frac{1}{\sigma }\times \frac{L}{S}$$
where *L* is the distance between two contact points of probe and ZnO nanowire, *S* is the cross-sectional area of ZnO nanowires and *R* is resistance of ZnO nanowires.

### 2.2. Piezoelectric Properties Characterization Principles

ZnO nanomaterials possess piezoelectric properties. Nanogenerator made of piezoelectric nanomaterials has become a hot research topic [23,24,25]. Principle of piezoelectric properties measurement, i.e., the two-probe strategy, is shown in Figure 2.

Two tungsten probes would be connected to sample to form an electrical circuit. One probe holds the nanowire, another one is used to deform the nanowire. As shown in Figure 2, when the probe is away from the sample, it would be electrically neutral. When the probe gradually moves to the right and touches ZnO nanomaterial, it exerts a force on the ZnO nanomaterial, which will deform. As ZnO nanomaterial have piezoelectric properties, deformation can generate piezoelectric electric field inside nanomaterial, thus generating a piezoelectric potential. The potential will be correlated with the deformation extent [26,27,28,29,30].

The piezoelectric constant is a significant parameter that measures conversion of mechanical energy to electrical energy by ZnO. We defined a shear piezoelectric constant to estimate piezoelectric capability of ZnO nanowire: $d_{shear}=\frac{Q_{shear}}{F}$, where *Q*_shear_ is the electric charge generated on two ends of the deformed nanowire, *F* is the force deform ZnO nanowire which is exerted by probe. 

### 2.3. Contact Strategy between Probe and the Nanorod

Object-probe contact strategy plays an important role in the experiment. For characterization, probe gets close to and then touch the sample. Probe pose is key to improve experimental efficiency and yield, as probe angle will affect operation efficiency, as shown in Figure 3. 

It is known that at microscale, van der Waals force is proportional to contact area, and in vacuum, van der Waals force between any two objects is generally attractive. Figure 3 shows three types of contact strategies between the tip and the sample. Probe 1 touches the sample with a point, probe 2 has an acute angle with the sample, while probe 3 is nearly parallel with the sample. The van der Waals forces of probe 1 and probe 3 are the lowest and highest. In electrical properties characterization, probe touches the sample and both get into contact, leaves the sample after measurement is over, and contact strategy 1 and 2 should be adopted. For sample pickup, probe and sample get into contact, the probe takes the sample away, so contact strategy 3 should be used, to strength adhesion force between the two.

## 3. Materials and Equipment

Experimental setup is shown in Figure 4. The nanomanipulator (Canada/Toronto Nano Instrumentation Inc, NB202-SEM) and AFM (Canada/Toronto Nano Instrumentation Inc, EM-AFM) can be installed in the vacuum chamber of SEM (SU3500, Hitachi, Japan). The manipulator has a coarse and fine three-degree-of-freedom (DOF) mode that can be switched in real time according to operational tasks. The probe was ST-20-0.5 from GGB Industries Inc, with a tip diameter of 0.5 μm and taper of 10°. Original ZnO nanorod (diameter: 100–2000 nm, length: 3–10 μm) and nanowire (diameter: 50–120 nm, length: 2–20 μm) powder was purchased from Nanjing XFNANO Materials Tech Co., Ltd, China.

For mechanical/electrical properties characterization of ZnO nanorods, 40 mg ZnO nanorod powder were mixed with 2 ml alcohol. A pipette was used to drop the solution onto a silicon wafer, which is attached to the corresponding sample stage after dried. For piezoelectric test, a tiny amount of nanowire powder was glued on a small piece of carbon tape, which is placed on one arm of the nanomanipulator, and characterized with the other two tips.

According to abovementioned working principles, the mechanical/electrical/piezoelectric properties were measured and recorded.

## 4. Results and Discussions

Figure 5 shows the random distribution of observed nanorod and nanowire on the substrate. It can be seen from the figure that samples randomly distributed on the substrate. Some samples gather or pile up in three-dimensional space, resulting in aggregation, as red-highlighted in Figure 5. Correspondingly, samples cannot be directly contacted and interacted, i.e., not all samples are suitable for operation and characterization. Only clear and single nanorod/nanowire can be picked. Criteria of suitable ZnO nanorod/nanowire for mechanical/electrical characterization are defined as follows: it must exist independently without winding, stacking and contacting; nanomaterial and silicon wafer should be kept clean and without attachments.

SEM image scanning lacks operational depth information feedback. In experiments, probe manipulation relies on experience. In this paper, local scanning method is adopted to ensure high reliability of real-time visual feedback and accuracy of absolute probe positioning. Height information of probe and sample is confirmed by focusing. The sample was selected as reference and the probe was moved to touch sample by constantly adjusting nanomanipulator arm.

### 4.1. Electron Beam Irradiation

The point that a single probe contact with nanomaterial is called electron beam irradiation point, as shown in Figure 4. Electron beam irradiation is also one of the factors affecting contact between probe and its object [31]. As the probe tip and ZnO nanomaterial were in contact, their adhesion force increases with the electron beam irradiation time, whose increase rate depends on electron beam current, in accordance with EBID (electron beam-induced deposition). Even after suspending the irradiation, the adhesion force caused by electron beam irradiation increases irreversibly. 

Experiment has proved that possibility of samples attaching to the tip increases with operation time, as shown in Figure 6, so acceleration voltage of SEM and contact duration between probe and sample should be defined tactfully.

As seen from the Figure 6, when contact strategy 3 in Figure 3 was used, the nanowire-probe contact area is considerable, when irradiation time exceeds 240 s, ZnO nanowire was almost adsorbed to the tip irreversibly. For piezoelectricity characterization, probe need to pick up a single ZnO nanowire from tape which adsorb a large number of nanowires. Accordingly, sample-probe contact duration should last four minutes at least.

### 4.2. Morphology and Mechanical Properties of ZnO Nanorods

Mechanical properties of ZnO nanorods have a very important impact on their applications in micro/nano devices [32,33,34]. Therefore, it is of great significance to characterize the mechanical properties of ZnO nanorods. Surface topography of a ZnO nanorod is shown in Figure 7a,b. Combining with SEM, it is easy to locate ZnO nanorod, and characterize it with AFM efficiently.

For selected ZnO nanorod in this experiment, the average Young’s modulus and spring rate of ZnO nanorods are 1.40 MPa and 0.08 N/m, respectively.

### 4.3. Electrical Properties of ZnO Nanorods

Accelerating voltage was adjusted to 30 kV when probes contact with ZnO nanorod, where strategy 2 was used, and irradiation time was set 15 minutes, which was used to strengthen adhesion between probe and nanorod. This operation can effectively reduce contact resistance between probe and nanorods. Contact interaction is shown in Figure 4.

Contact between probe tip and the end surface of ZnO nanorods is a typical metal semiconductor contact. Through nanomanipulator embedded in SEM, it is connected with external electrical test module to form a current circuit. The I-V characteristics of the probe tip and the contact barrier are measured and evaluated. Three groups of ZnO nanorods with different lengths were tested, as shown in Figure 8.

When the Schottky barrier is in positive or reverse bias state, the depletion layer in ZnO nanorods increases with the gradual increase of bias voltage. When the critical bias is reached, the conducting channels in the nanorods will be completely exhausted. This depletion behavior is similar to clamping of conductive channels in a field effect transistor. And the current will show a cut-off characteristic after clamping. The experiment shows that the current will not change when the forward voltage reaches 4.5 V and the reverse voltage −3.8 V. The results show that ZnO nanorods and tungsten probes form a good back-to-back Schottky contact and show a cut-off characteristic.

### 4.4. Electrical Conductivity of ZnO Nanowires

Electrical conductivity of ZnO nanowires are measured and evaluated by nanomanipulator embedded in SEM. Three groups of ZnO nanowires with different lengths were tested, as shown in Figure 9.

As can be seen in Figure 9d, the blue and orange bars represent the resistance and electrical conductivity of ZnO nanowires, respectively. Resistance increases with measured length, consistent with theoretical predictions. But the electrical conductivity also increased with the increase of measured length, although the change is not significant. This may be due to the poor heat dissipation in the vacuum chamber of SEM. The current equipment can not completely exclude the influence of temperature, which can be improved by setting thermocouple inside SEM or other ways in future research. Finally, the average electrical conductivity of ZnO nanowires is calculated as 480.9 S/m.

### 4.5. Piezoelectric Properties of ZnO Nanomaterials

Due to the small size and ability to supply energy to micro/nano devices, metal–oxide-based piezoelectric energy harvesters have recently gained an enormous amount of attention. Here piezoelectric properties of a single ZnO nanowire is measured by nanomanipulator embedded in SEM with a two-tip strategy. Experimental apparatus is shown in the Figure 4, and experimental process are shown in Figure 10a–d.

The two-tip strategy is that, a clear, single ZnO nanowire on the carbon tape was selected, probe Ⅱ got close to the nanowire below its tip, as the nanowire moved slightly with the probe, the two got into contact. The probe Ⅱ position was adjusted to increase contact area until it exceeds 2 μm. The contact was exposed with e-beam for at least 4 min. According to Figure 6, probe Ⅱ and nanowire will be fixed together firmly. As diameter of ZnO nanowire is lower than that of ZnO nanorod, it is easier to separate from tape, increasing experiment efficiency.

The nanowire was then taken away from the tape by probe Ⅱ. Probe Ⅰ got into contact with free end the nanowire with the same strategy with probe Ⅱ. E-beam is turned off to eliminate electromagnetic noise. Probe Ⅰ went downward a specific distance with the nanowire free end, called deformation, the transient maximum voltage was recorded, as shown in Figure 10. As deformation increases, the voltage will also increase.

Deformation ranges from 0 to 6 µm. Accordingly, the output voltage maximum varies from 0 to 0.74 mV, in accordance with previous literature [12]. The shear piezoelectric constant is calculated to be around 9.91 × 10^−8^ C/N [35]. It is much higher than previous literature (26 pC/N) [36]. The discrepancy may be results of different crystal direction or deformation range. More accurate and reasonable piezoelectric constant will be our future work, with the benefit of more accurate microforce sensors. More accurate and reasonable piezoelectric constant will be our future work, with the benefit of more accurate microforce sensors. 

Our work provides theoretical basis for the application of ZnO nanowires, such as nanogenerators. The two-tip strategy is simple and efficient, and the probes are free from abrasion. Improving its repeatability, adaptability and reliability is our future work.

## 5. Conclusions

Precise operation/characterization of nanomaterials is still one of the most important challenges in the construction of micro/nano scale components. Mechanical/electrical properties characterization of ZnO nano-materials relies on nanomanipulator and AFM embedded in SEM. Surface morphology and mechanical properties were observed by AFM. Electrical properties were characterized with nanomanipulator, which showed that the ZnO nanomaterial have cut-off characteristics and good schottky contact with the tungsten probes. Piezoelectric property was also measured, and the two-tip strategy was convenient, efficient and easy to expand to other materials. Our work provides useful references for nanomaterial characterization and also theoretical basis for nanomaterials application.

## Figures and Tables

**Figure 1 micromachines-12-00248-f001:**
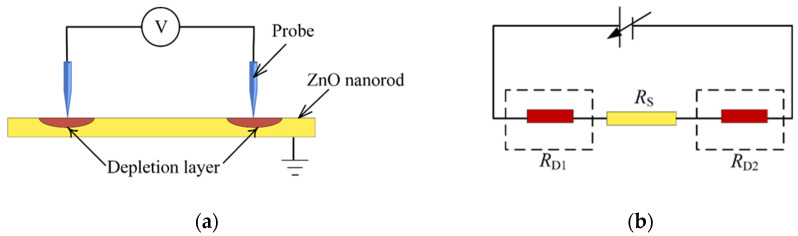
Equivalent electric circuit of metal-semiconductor-metal structure. (**a**) Probe is marked in blue. When semiconductor ZnO nanorod come into contact with probe electrode, a barrier is formed at the contact point which is called depletion layer. (**b**) Three resistors in the circuit. $R_{D1}$ and $R_{D2}$ are resistance of the depletion layer, $R_{s}$ is resistance of the ZnO nanorod.

**Figure 2 micromachines-12-00248-f002:**
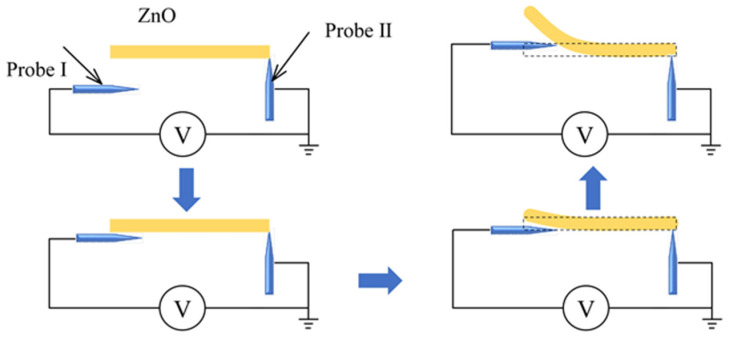
Principle of piezoelectric properties characterization. Yellow rectangle represents a single ZnO nanowire. ZnO nanowire deforms when pressed by probe which is marked in blue. Two probes contact ZnO nanowire to form an electrical circuit. A digital multimeter measures voltage generated by ZnO nanowire deformation.

**Figure 3 micromachines-12-00248-f003:**
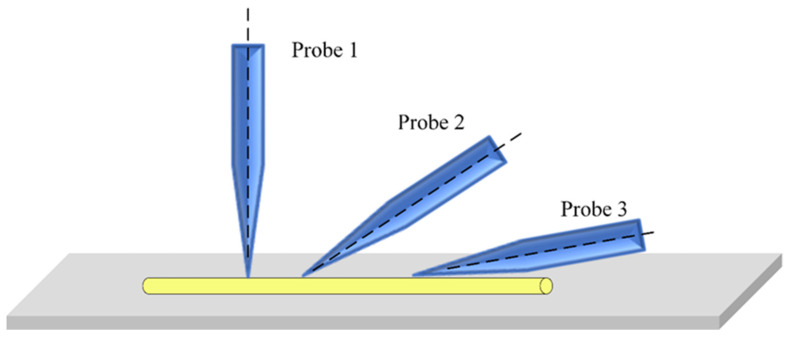
Force interaction between probe and nanorods with different interaction angles.

**Figure 4 micromachines-12-00248-f004:**
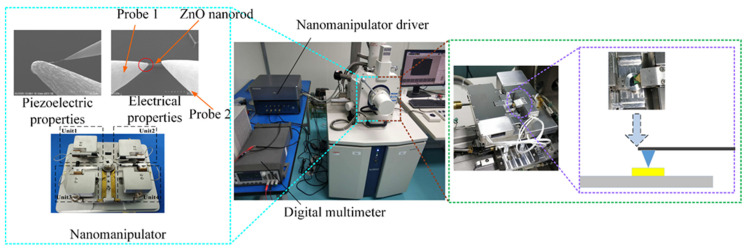
Mechanical/electrical characterization experimental apparatus. On the left is the electrical/piezoelectric characterization system (The point that a probe contacts ZnO nanorods is called irradiation point, as red dotted line marks), in the middle is the SEM, and on the right is the morphology and mechanical characterization system.

**Figure 5 micromachines-12-00248-f005:**
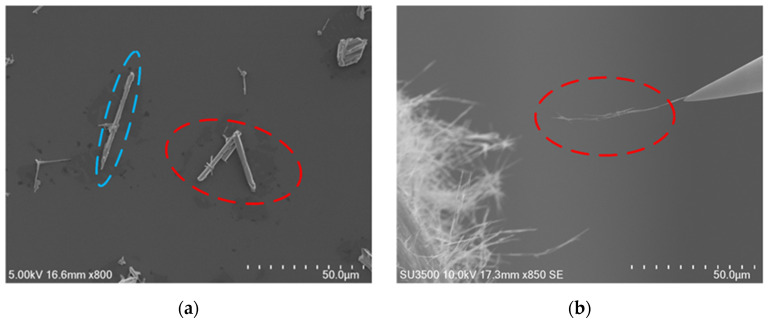
ZnO nanorods and ZnO nanowires. (**a**) ZnO nanorod marked by blue dotted line is suitable for operation and observation and marked by red dotted line means stacked ZnO nanorods. (**b**) ZnO nanowires marked by red dotted line are sample for piezoelectric properties characterization.

**Figure 6 micromachines-12-00248-f006:**
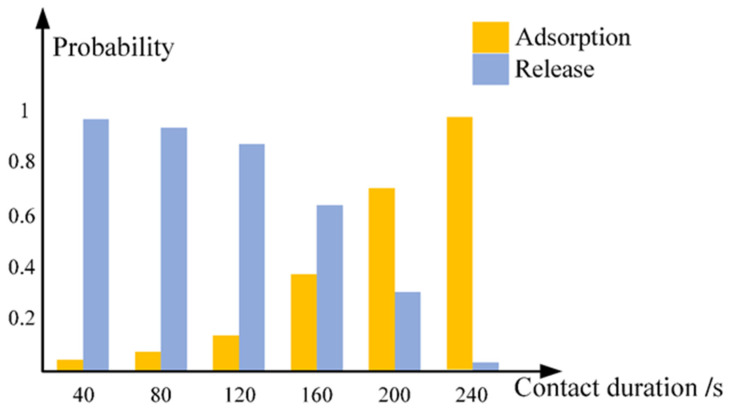
Release/adsorption probability vs. irradiation time between probe and nanorod. Contact strategy 3 was used. SEM acceleration voltage is 5 kV. Adsorption probability reached over 95% when contact time exceeds 4 minutes.

**Figure 7 micromachines-12-00248-f007:**
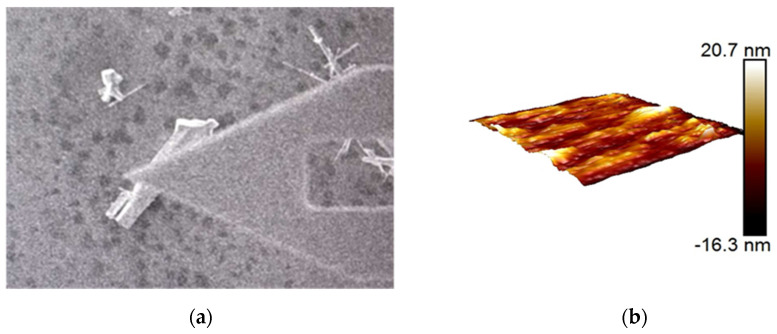
Morphology and mechanical properties of ZnO nanorods in the range of 2 µm × 2 µm. (**a**) AFM is scanning a single ZnO nanorod. (**b**) Morphology of an observed nanorod.

**Figure 8 micromachines-12-00248-f008:**
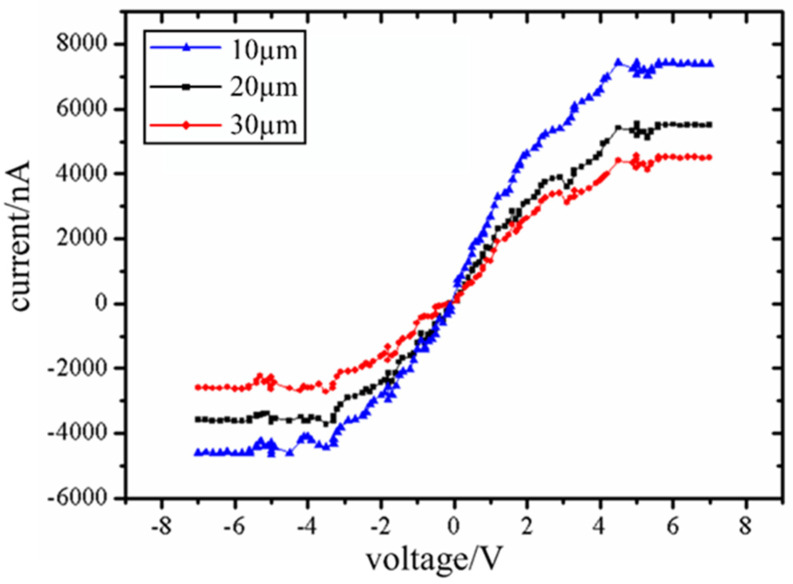
I-V curve of three groups of nanorods. Curve 1, curve 2 and curve 3 correspond to ZnO nanorods with length of 10 µm, 20 µm and 30 µm respectively, with same diameters of 2 µm. The three I-V curves are nearly symmetrical, which indicates that ZnO nanorods and probe form almost equal contact barriers between the forward bias junction and reverse bias junction.

**Figure 9 micromachines-12-00248-f009:**
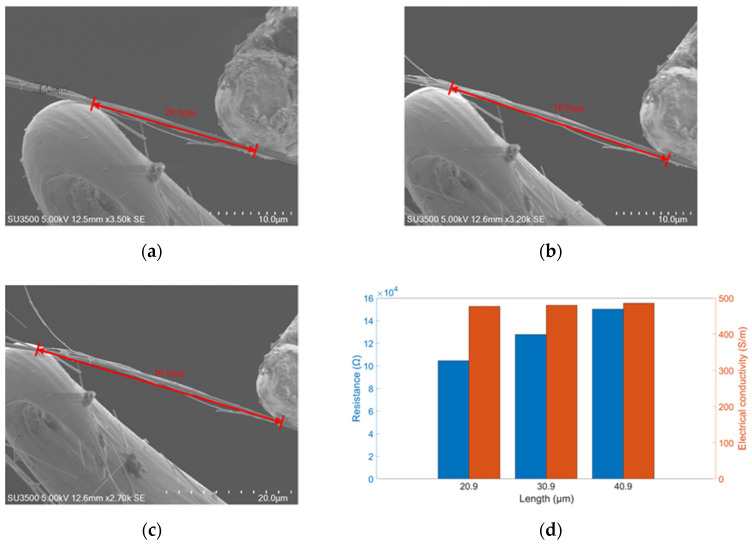
Electrical conductivity measurement. (**a**–**c**) Lengths of ZnO nanowires are 20.9, 30.9, 40.9 µm with same diameters of 1.06 µm. (**d**) Blue bars represent resistance and red bars represent electrical conductivity correspond to ZnO nanowires with length of 20.9 µm, 30.9 µm and 40.9 µm.

**Figure 10 micromachines-12-00248-f010:**
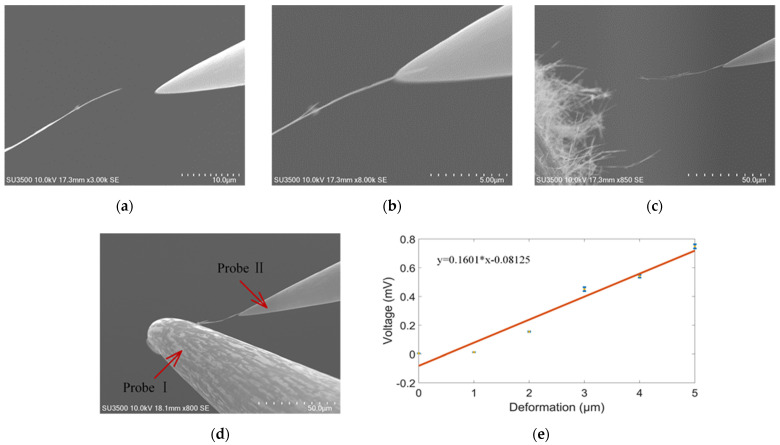
Piezoelectric characteristic measurement. (**a**) ProbeⅡgets close to ZnO nanowire. (**b**) ProbeⅡtouches and picks up ZnO nanowire. (**c**) ProbeⅡtakes the nanowire away from the tape. (**d**) ProbeⅠdeforms the nanowire. (**e**) Piezoelectric voltage vs. deformation. Deformation ranges from 0 to 6 µm. Accordingly, the output voltage varies from 0 to 0.74 mV. *n* = 15.

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
