# Peer review of "Mechanical/Electrical Characterization of ZnO Nanomaterial Based on AFM/Nanomanipulator Embedded in SEM"

_micromachines, 2021, doi:10.3390/mi12030248_

Round 1
Reviewer 1 Report
the title of the paper (Mechanical/electrical characterization of ZnO nanomaterial
3 based on AFM/nanomanipulator embedded in SEM) suggests that numerical values of mechanical and electrical properies of piezoelectric ZnO NWs will be given.
in rows 184 and 185 the authors write:
The average elastic constant and rigidity of ZnO nanorods are 1.40 MPa and 0.08 N/m, respectively.
is rigidity the shear modulus? why N/m and not N/m2?
what do you mean for elastic constants?
Young modulus, shear modulus?
please specify clearly what have been measured and use the correct unit (N/m2).
can the authors quantify the piezoelectric constants of ZnO?
can the authors quantify the electrical conductivity of ZnO?
Reviewer 2 Report
In this manuscript, the authors investigated the electric and piezoelectric property of ZnO nanorods (I-V curves of those with various lengths) and nanowires, respectively. It is important to describe the detailed experimental techniques (especially, the adsorption of the nanowire by electron beam irradiation). Thus, the manuscript is publishable in this journal.
However, there are some minor problems and questions as follows.
Figure 9d shows the SEM image in the piezoelectric properties characterization, which is represented also in Figure 2. However, the real image may hardly correspond to the schematic images. It may be better to revise Figure 2 or add the schematic figure for Figure 9d. Is the "deformation", which seems to be the moving distance of the probe 2, the same or corresponding length of the transversal strain as shown in Figure 2? The correspondence may be established only with the same configuration of the probe 1 and 2 as Figure 9d.
Line 176: "absorbed" -> "adsorbed"
Round 2
Reviewer 1 Report
the piezoelectric constants are measured in m/V or C/N: why the author measure it in C/m? how can I compare their result with that from others (for example: B. Christian et al., Piezo-force and vibration analysis of ZnO nanowire arrays for sensor application, Procedia Engineering 168 (2016) 1192 – 1195, where ZnO NWs piezoelectric constant d33 = 15 pm/V has been determined by using three methods)? please use the correct units and justify your revision.
Round 3
Reviewer 1 Report
.